# The use of quarantine as an international travel measure during the COVID-19 pandemic: A comparative analysis of implementation and equity impacts in five "exemplar" countries

Kelley Lee[1]*, Jane Williams[2], Yi-Chin Wu[3], Aysha Farwin[4], Youmi Kim[5], Sungkyu Lee[6], Natasha Howard[4], Salta Zhumatova[1]

**1** Pandemics and Borders Project, Faculty of Health Sciences, Simon Fraser University, Burnaby, British Columbia, Canada, **2** School of Public Health, University of Sydney, New South Wales, Australia, **3** Center for Global Health Research, Foundation of Medical Professionals Alliance in Taiwan, Taipei, Taiwan, **4** Saw Swee Hock School of Public Health, National University of Singapore and National University Health System, Singapore, Singapore, **5** Division of Emergency Preparedness and Response, Korea Disease Control and Prevention Agency, Seoul, Republic of Korea, **6** Korea Center for Tobacco Control Research and Education, Seoul, Republic of Korea

* kelley_lee@sfu.ca

## Abstract

During the COVID-19 pandemic, virtually all countries used a range of measures to mitigate the risks of virus introduction and onward transmission via international travel. Quarantine was a key international travel measure (ITM) used by governments to achieve public health goals. However, the highly varied ways in which quarantine was implemented makes lesson learning for future pandemics challenging. Moreover, most studies overlook the secondary impacts of ITMs on individuals and populations including potential inequities in their distribution. This paper comparatively analyses five countries deemed exemplary in their implementation of quarantine during COVID-19. Building on Damschroder's Consolidated Framework for Implementation Research, we apply eight variables to identify similarities and differences in quarantine use. We drew on our standardized coding of ITMs in the WHO Public Health and Social Measures dataset, and additional on-line searches, to compile data on each variable. Findings show that the five countries were early adopters of quarantine, applied them relatively stringently, and maintained them throughout the emergency phase of the pandemic to effectively advance public health goals. However, the countries differed in how secondary impacts were managed, resulting in the inequitable distribution of opportunity and burden for some individuals and populations. We conclude that exemplary implementation of quarantine during future public health emergencies should consider both public health goals and the equity of secondary impacts.

**Data availability statement:** All sources of data used in this paper, compiled from publicly available sources, are referenced in the manuscript. This includes access to the full WHO Public Health and Social Measures dataset which is made available by the London School of Hygiene and Tropical Medicine at: https://github.com/lshtm-gis/WHO-PHSM.

**Funding:** KL received funding for the Pandemics and Borders Project from the Canadian Institutes of Health Research (Grant Number CIHR 178276) and the New Frontiers in Research Fund (Grant Number NFRFR-2022–00241). SZ was funded by the Pandemics and Borders Project. The funders had no role in study design, data collection and analysis, decision to publish, or preparation of the manuscript.

**Competing interests:** The authors have declared that no competing interests exist.

## Introduction

International travel measures (ITMs), defined as a policy or intervention applied for the purpose of managing human mobility between two or more countries, were applied by virtually all governments during the COVID-19 pandemic. With limited precedence and available evidence to inform their use during a large-scale and prolonged public health emergency, governments applied ITMs in highly varied ways (e.g., types of measures, timing, duration, scope and stringency). The uncoordinated, frequently changing, and poorly evidenced ways that travel measures were used by most governments proved "chaotic" [1] for public health officials, travellers, and the travel sector. While initially supported in most countries, ITMs over time became a major source of public controversy.

A key type of ITM used during COVID-19 was quarantine defined as the separation of a person or group who may have been exposed to a notifiable pathogen posing a risk to a wider population. *Isolation* refers to the separation of a person or group who is confirmed as infected with such a pathogen. Both practices have been used as a public health measure since at least the fourteenth century to control the spread of disease [2]. The use of either quarantine or isolation as an ITM during the COVID-19 pandemic was colloquially referred to as quarantine. Governments routinely apply quarantine at international points of entry for inbound and outbound traffic and trade [3,4]. Based on the WHO Public Health and Social Measures dataset, at least 110 countries used quarantine as an ITM between January 2020 and May 2023 [5].

The assessment of quarantine as an ITM has to date been limited in three key respects. First, studies have heavily focused on travel restrictions, as one type of ITM, defined as the control of who can be an inbound and/or outbound traveller. While travel restrictions were primarily used during the initial phase of the pandemic, governments subsequently applied other types of ITMs instead of, or in addition to, restrictions. These included screening (notably testing), immunity certification (notably vaccination status), and quarantine [6]. There remains an important need to assess different types of ITMs used during COVID-19, applied singly and in combination, to inform their effective use in future public health emergencies of international concern (PHEIC) [7].

Second, studies have so far focused on assessing the effectiveness of ITMs at advancing public health goals, namely pathogen introduction and onward transmission [8–10]. However, systematic reviews of the evolving evidence report that the secondary impacts or unintended outcomes from ITMs remain understudied [11]. This is confirmed by scoping reviews of the economic [12] and social [13,14].impacts of ITMs. The shift by the World Health Organization (WHO) during the COVID-19 pandemic, from blanket recommendations against travel restrictions during PHEICs, under States Parties' commitments under the International Health Regulations (IHR), to supporting risk-based approaches to ITMs [15], requires fuller understanding of their public health and broader societal impacts. The latter includes attention to equity considerations, and whether ITMs disproportionately impacted some individuals and populations over others.

Third, evidence reviews suggest that quarantine, in general, was an effective intervention for reducing virus introduction and onward transmission. However, this evidence summarized a broad range of national settings which varied substantially in how quarantine was implemented. As Abou-Setta et al. write, "it was not clear in most studies if quarantine was mandatory, how it was enforced, and the consequences of refusal of the intervention. This is a limitation of the implementation of the intervention and the reporting of the studies." Grepin et al. conclude in their narrative synthesis of five published systematic reviews that "on balance, some border control measures were more effective than others, in particular quarantine, but a great deal about the context and implementation of these measures remains poorly understood." [11].

This paper addresses the above knowledge gaps by comparatively analysing the context and implementation of quarantine during COVID-19 in five exemplar countries: Australia, Aotearoa New Zealand, Singapore, South Korea, and Taiwan. These countries were praised by policy makers, scientists, media and the public for their low incidence of travel-related virus introduction and onward transmission. Fuller comparison of the policies and practices of these exemplar countries, and the impact of quarantine on both public health and other policy goals, would help to inform the future use of quarantine as an ITM during PHEICs.

We begin by developing and applying a heuristic framework. We then apply a heuristic framework for describing and comparatively analysing each exemplar country's quarantine policy during COVID-19 encompassing legal framework and authority, timing, duration, eligibility, location, cost, accessibility, and conditions. While all five countries are deemed exemplars, we apply an equity lens focused on distributive justice to better understand the secondary impacts of quarantine use. This includes any populations that may be disproportionately impacted. We conclude by drawing lessons for integrating equity considerations into quarantine use as ITMs in future pandemics. We argue that increased attention to equity is essential for public trust and compliance with ITMs, and as a core value in what is deemed exemplary public health policy.

## Approach and methods

### Research statement

As this study does not involve human subjects, it is exempt from requiring research ethics board approval. The Pandemics and Borders Project has received ethics approval under the Simon Fraser University Research Ethics Board (Study Number: 30000604).

### Primary and secondary data collection

This study collected and analysed primary and secondary data from publicly available sources. We began with the WHO Public Health and Social Measures (PHSM) dataset, a repository of COVID-19 interventions adopted by WHO member states. This dataset was compiled by WHO in collaboration with the London School of Hygiene and Tropical Medicine and other organizations [16] which scraped on-line data from primary and secondary sources: a) government websites; b) WHO country offices; c) direct reporting by WHO member states; d) international and national media; and e) thematic webpages on PHSM. Data were classified by the PHSM dataset into seven categories and 42 subcategories of measures [17].

Given the varied implementation of travel measures worldwide, in our review of this dataset, we found terminology inconsistent and sometimes imprecise. To enable comparative analysis, we developed a standardized taxonomy of COVID-19 ITMs to recode the PHSM dataset [18]. Through recoding, we identified around 110 countries that used quarantine as an ITM during the COVID-19 pandemic.

### Case study selection

Using our recorded PHSM dataset, we identified five countries deemed to be exemplary in their use of quarantine. Exemplar is broadly defined as the effective use of quarantine measures to reduce international travel-related SARS-CoV-2 introductions and onward transmission. The countries were selected for being praised in the following sources:

- Mainstream media [19,20];

- Public health and government officials [21];

- Scholarly literature [22–26];

- High ranking on international travel controls on the OxCGRT Project *Containment and Health Index* [27] and *Stringency Index* [28]; and

- The Exemplars in Global Health Project (Box 1) [29].

---

**Box 1  Examples of how the case studies were described as exemplar.**

**Australia**

The quarantine system "has proven largely successful in protecting Australia from many of the impacts of COVID-19 by controlling the entry of COVID-19 at the border and preventing its spread into the Australian community" [30]. A report by the Australian National Audit Office concluded that "The management of human biosecurity for international air travellers during COVID-19 [which included mandatory quarantine and quarantine-free flights] has been largely effective" [31].

**Aotearoa New Zealand**

"Effective pandemic tools matter. The pandemic response has required New Zealand to swiftly develop a new set of tools for managing this threat. They include systems for border management and quarantine…" [20]

**Taiwan**

"Taiwan adopted strict strategies such as border controls, quarantine, and other community-level measures, resulting in a remarkable 253-day streak without local transmissions; earning unprece-dented international recognition" [32].

**Singapore**

"Stay-Home Notice (SHN) regime was introduced on 17 February 2020 to quarantine travellers and "ringfence" risk while limiting economic impact. Our early contact-tracing efforts successfully contained a wide transmission [33].

**South Korea**

"South Korea isolated infected patients, increased compliance by supporting those in quarantine, and traced contacts with unusual thoroughness" [34].

---

For these five countries, we then extracted data on their quarantine use from January 1, 2020, to May 31, 2023. We supplemented PHSM data with secondary data obtained by on-line searches of government websites, news outlets, and other public websites. English-language sources were primarily used, with Google Translate used for Mandarin-language websites. Additionally, we incorporated data from web-based crisis management systems, such as Global Monitoring, which track, analyze, and communicate critical events globally.

**Framework for comparative analysis of travel measure implementation**

To comparatively analyse the implementation of quarantine by the five countries during COVID-19, we developed a framework based on Damschroder's Consolidated Framework for Implementation Research [35]. This framework analyzes the implementation process by examining several core domains:

- Intervention: The core characteristics of the planned policy itself.

- Inner Setting: The context within the organization or specific setting where the policy is being implemented.

- Outer Setting: The external political, economic, and social contexts influencing implementation.

- Individuals Involved: The characteristics of the people or groups involved in implementing the policy.

- Process: The active steps and strategies used to implement the policy.

Applying this framework, we identified eight key variables for which data was publicly accessible (Table 1). Data collected from primary and secondary sources were compiled for each variable and are made available in the on-line supplementary annexes.

In addition, we used the recorded PHSM dataset to created a panel dataset to understand how the five countries implemented quarantine over time. Months and years serve as the temporal units of analysis for when quarantine measures were implemented. The dataset includes variables for country ID and date, a quarantine policy variable measured on a 0–1 scale, and nominal categorical variables identifying the countries from which travellers were subject to quarantine, as well as categories of travellers targeted by quarantine. Data collection sought to document quarantine policies for each month of the pandemic, with the selection criteria based on references to policies or policy documents cited in the sources. This dataset was used to generate summary graphs of each country's quarantine use.

**Approach to assessing equity in quarantine use**

Quarantine by definition is a restrictive measure that seeks to limit the freedoms of selected individuals for the benefit of a wider community. Given that quarantine can be highly burdensome and liberty limiting, ethical issues are raised. For example, can the use of quarantine be justified as a proportionate response to the risk posed? Is the burden placed on people subject to quarantine warranted by the expected benefit to society? Are people who undergo quarantine minimally

**Table 1. Framework to compare quarantine implementation.**

| Variables | Questions |
|---|---|
| Legal authority | What type of political system was in place?<br>What law(s) provided the authority to use quarantine?<br>What was the stated purpose of the legislation?<br>What was the strength and scope of authority provided under the legislation? |
| Timing | When was quarantine introduced?<br>When was quarantine lifted?<br>Was quarantine applied continuously or intermittently?<br>How often did quarantine requirements change? |
| Duration | What was the length of time a person needed to quarantine? |
| Persons/populations subject | What individuals or groups were subject or not to quarantine? |
| Location | Where were individuals permitted or required to quarantine? |
| Cost | How much did it cost to quarantine?<br>Who paid for the cost of quarantine? |
| Access | How did individuals gain access to designated quarantine sites (e.g., hotel)? |
| Conditions | What food was provided and of what quality?<br>What was the level of amenities and facilities?<br>Was there access to appropriate healthcare?<br>Were special needs provided for (e.g., accessibility)?<br>Were the needs of priority populations provided for (e.g., children, older adults)? |

burdened or harmed? Should individuals be compensated for burdens incurred and, if so, to what extent? Are there alternative and equally effective public health measures available that are less restrictive (e.g., vaccines)? [36] These types of ethical considerations underpin States Parties commitments under IHR Article 32, on the treatment of travellers when using ITMs including quarantine, although these principles remain broadly stated (Box 2).

---

**Box 2  International Health Regulations (2005), Article 32 Treatment of travellers.**

In implementing health measures under these Regulations, States Parties shall treat travellers with respect for their dignity, human rights and fundamental freedoms and minimize any discomfort or distress associated with such measures, including by:

(a) treating all travellers with courtesy and respect;

(b) taking into consideration the gender, sociocultural, ethnic or religious concerns of travellers; and

(c) providing or arranging for adequate food and water, appropriate accommodation and clothing, protection for baggage and other possessions, appropriate medical treatment, means of necessary communication if possible in a language that they can understand and other appropriate assistance for travellers who are quarantined, isolated or subject to medical examinations or other procedures for public health purposes.

Note: This article was unchanged by revisions adopted in May 2024.

Source: https://apps.who.int/gb/wgihr/pdf_files/wgihr8/WGIHR8_Proposed_Bureau_text-en.pdf

---

WHO defines equity as "the absence of unfair, avoidable or remediable differences among groups of people, whether those groups are defined socially, economically, demographically, or geographically or by other dimensions of inequality (e.g., sex, gender, ethnicity, disability, or sexual orientation)" [37]. Building on this definition, we draw upon justice theory to advance assessment of equity in quarantine implementation as an ITM. Political philosophy and public health ethics posit that there are four main types of justice - procedural, distributive, retributive and restorative [38].

*Procedural justice* concerns the fairness by which processes are conducted, for example, in priority setting, resource allocation, enforcing rules, and resolving disputes. For quarantine, this might concern how measures were set (i.e., did decisions align with appropriate authority, what groups were included/excluded, were conflicts of interest managed well, were decisions transparent); implemented (i.e., were measures clear and understandable, were people treated with dignity and respect) and enforced (i.e., was there a fair dispute resolution process). Given limited publicly available data on the processes related to quarantine use in the five countries, we are unable to assess equity from a procedural justice perspective in a detailed way. However, we can make some broader assessments based on legal mandate and timing. *Retributive justice* concerns fairness in the assignment of punishments or penalties to wrongdoers based on the consequences of culpable actions [39]. For quarantine, this might concern the appropriate penalties (e.g., for those noncompliant with quarantine requirements based, for instance, on resultant illness and death caused, or costs to the healthcare system. *Restorative justice* focuses on repairing the harm done by the perpetrator and rebuilding that person's relationship with the victim and society [40]. For quarantine, this might concern community service or reparations to benefit those harmed by non-compliance. While quarantine was legally mandated for some travellers during COVID-19 in each of the five countries, the high degree of compliance precludes analysis of retributive and restorative justice.

PLOS Global Public Health

Given the above, this paper will focus on *distributional justice* which concerns the idea of fairness in the allocation of benefits and burdens from an activity in a society. Quarantine should not be systematically more burdensome or difficult for some groups to discharge than others. For quarantine measures during COVID-19, first, we sought to understand the *distribution of opportunity*. In countries where quarantine took place in supervised locations, how was quarantine accommodation managed, especially when demand exceeded supply? To what extent was this scarce resource managed equitably? In countries where quarantine could be served in a place of residence, how equitably was the opportunity to do so for different population groups? We analysed who was subject, access, and location in the five countries to understand the systematic distribution of quarantine burden.

Second, we considered equity by the *distribution of burden*. This concerns how the benefits and costs of quarantine were distributed among individuals and different populations. Were quarantine arrangements disproportionately beneficial or burdensome for some populations? Did the cost of quarantine compliance disproportionately impact any populations? Were basic needs equitably provided for? Were opportunities to minimize the burden from quarantine equitably distributed among population groups? Were the secondary costs of quarantine, such as lost employment or educational opportunities, experienced equitably among all populations? We analysed who was subject to quarantine, its cost, duration, and conditions in the five countries to understand the systematic distribution of burden.

## Results

### Legal authority for quarantine use

The legal authority for the five countries to use quarantine as an ITM is set out in the WHO IHR and national legislation (S1 Annex). Four of the five countries are States Parties to the IHR while Taiwan, unrecognized as a WHO member state, is officially included in the implementation framework. WHO may issue recommendations to States Parties with respect to persons on the implementation of "quarantine or other health measures for suspect persons" [41]. States Parties may "compel the traveller to undergo or advise the traveller…to undergo…isolation, quarantine or placing the traveller under public health observation" if "there is evidence of an imminent public health risk." These measures would be applied "in accordance with its national law and to the extent necessary to control such a risk" [42]. As described in Box 2, Article 32 states that, in using such measures, States Parties "shall treat travellers with respect for their dignity, human rights and fundamental freedoms and minimize any discomfort or Distress" [43].

The national laws for quarantine use as an ITM during COVID-19, setting out when quarantine may be used, the extent of government authority, and the bodies responsible for implementation depended on the political structure of each country. Four of the countries have unitary systems whereby the central government holds authority over quarantine. Australia is the exception with the Biosecurity Act (2015) giving authority to the Commonwealth (Federal) Government to adopt human biosecurity emergency powers and manage international borders [44]. These powers were exercised in February 2020 with the 14-day quarantine of returning nationals on Christmas Island amid the emerging COVID-19 pandemic. However, subsequent implementation of quarantine for international arrivals was the responsibility of six state and two territory governments, supported by amendments to their Public Health Acts. In practice, the majority of quarantine implementation was borne by New South Wales as the main destination of most international arrivals. This devolved responsibility meant that quarantine was variably implemented across Australian jurisdictions.

South Korea was the only country of the five with a designated Quarantine Act [45]. After the MERS outbreak in 2015, the Infectious Disease Control and Prevention Act was revised to give increased decision making authority to the central government, the Ministry of Health and Welfare (MoHW) and Korean Centre for Disease Control (KCDC) during health emergencies. The law was further revised in March 2020 amid the emerging COVID-19 pandemic to include increased fines for noncompliance with quarantine requirements.

While quarantine is embedded within broader legislation in the other four countries, government authority to use quarantine was temporarily expanded in three countries during COVID-19. Singapore introduced the COVID-19 (Temporary Measures) Act in 2020 as an expansion of the Infectious Diseases Act (IDA) [46]. While the IDA required the notification

of specified infectious diseases, and allowed the Health Ministry to take necessary actions to prevent the importation of infectious diseases, it did not extend to the enforcement of control orders. The new legislation empowered the Minister to a) impose mobility restrictions, b) convict individuals who contravened a control order; and c) appoint enforcement officers to ensure compliance. This served as the legal basis for the strict use of quarantine as an ITM [47]. Similarly, Aotearoa New Zealand adopted the COVID-19 Public Health Response Act 2020 [48], and a series of Section 70 orders setting out new requirements for incoming travellers [49]. Overall, while there was legal authority to use quarantine in all five countries for infectious disease control, revisions to existing laws were needed to extend this authority for use as an ITM.

### Timing of adoption and duration of quarantine use

Evidence from COVID-19 suggests that early and stringent implementation of ITMs was more effective at achieving public health goals [50–52]. As described by Aotearoa New Zealand Director-General Ashley Bloomfield, "it's all about the speed. The faster you can find the cases, isolate the cases, and track their close contacts, the more successful you're going to be" [53]. All five countries were among the earliest adopters of quarantine (S2 Annex). Taiwan was the earliest, implementing "onboard quarantine of all direct flights arriving from Wuhan, China" on 1 January 2020 [54].

Timing also concerns how long quarantine measures were kept in place. Our analysis shows the five countries applied quarantine for some international travel for 74% (Australia and South Korea) to 100% (Aotearoa New Zealand) of the period between January 2020 and May 2023 (Fig 1). This compares with countries such as the US (28%) and Brazil (40%). However, evidence is mixed regarding the public health effectiveness of long-term quarantine use. Maintaining effective quarantine requires substantial resources, stringent enforcement and high compliance which can become challenging over time [55]. In some countries, such as the UK and Canada, these challenges led to quarantine requirements being introduced, lifted and reintroduced in response, for example, to variants of concern [56]. While the five countries studied maintained quarantine requirements throughout the pandemic, given the substantial secondary impacts, Singapore, Taiwan [57], Australia and Aotearoa New Zealand exempted certain categories of travellers using risk assessment based on variables such as source country, citizenship, purpose of travel, testing, and immunity status. However, efforts to create travel "bubbles" or "corridors" (within or through which ITMs were waived) proved short-lived when public health indicators worsened.

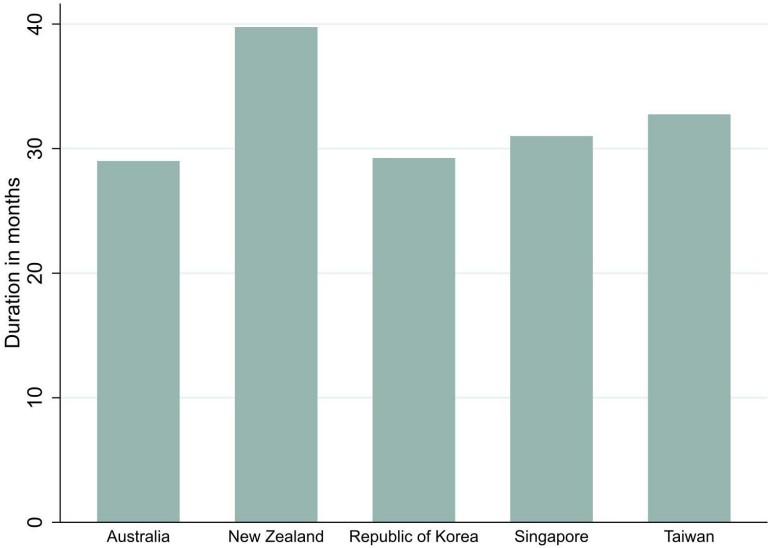

**Fig 1. Total months quarantine applied in the five countries (January 2020-May 2023).**

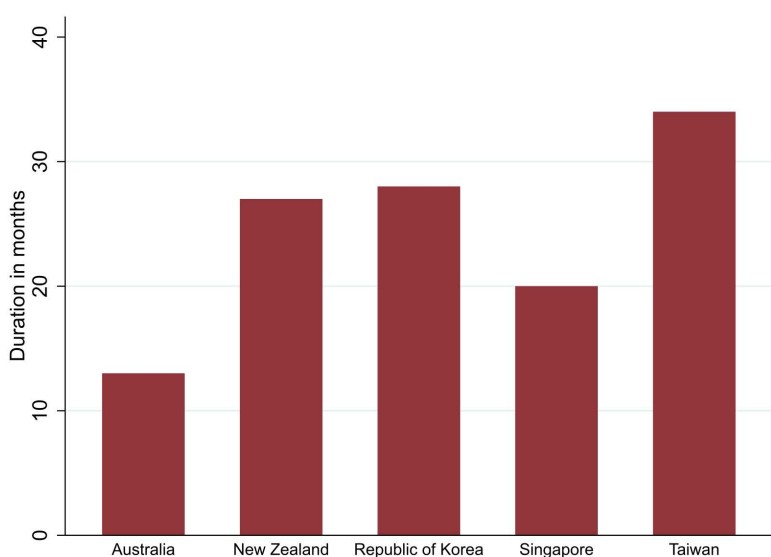

PLOS Global Public Health

## Length of quarantine period

The length of quarantine period required for international arrivals in the five countries varied over time (S3 Annex), ranging from 21 days in Singapore (7 May 2020-23 June 2020) to three days in Taiwan (15 June 2022-13 October 2022). In general, all five countries applied a longstanding base period of 14 days during 2020–2021, with Australia and Taiwan maintaining this policy for the longest period. Australia introduced a 14-day quarantine requirement on 29 January 2020, for evacuated nationals/residents arriving from China, until 21 February 2022 when some Australian states reduced the quarantine period to 7 days for unvaccinated inbound travellers. Taiwan applied a 14-day quarantine requirement from late January 2020 until March 2022, after which the period was gradually reduced from ten, three and then zero days by October 2022.

The variation in length of quarantine in some of the countries was based on evolving evidence about the infectious period for SARS-CoV-2. In addition, Singapore varied its quarantine period between 0–21 days using risk assessment based on, for instance, citizenship/residency, infection rates in source countries, purpose of travel, immunity status, and test results. Overall, the five countries differed in how they balanced scientific uncertainty about the transmission of a novel virus, and the secondary impacts of long quarantine periods.

## Persons/populations subject to quarantine

The five countries quarantined most or all inbound arrivals during the pandemic regardless of their citizenship or residence status (S4 Annex). Eventually entry was restricted to citizens and residents, during periods of the pandemic, given the limited availability of designated quarantine accommodation. Fig 2 illustrates the number of months that quarantine was applied to arrivals from all countries and all categories of travellers without exception. This ranged from Australia at around 13 months, until exemptions were made for travellers arriving from Aotearoa New Zealand, to Taiwan at around 34 months.

As the significant economic and social impacts of reduced travel began to be recognized, limited categories of international arrivals deemed lower risk were permitted reduced quarantine requirements based on travel history and itinerary; source country (or region); test results (pre-departure, point of arrival, or post-arrival); and immunity status (proof of full

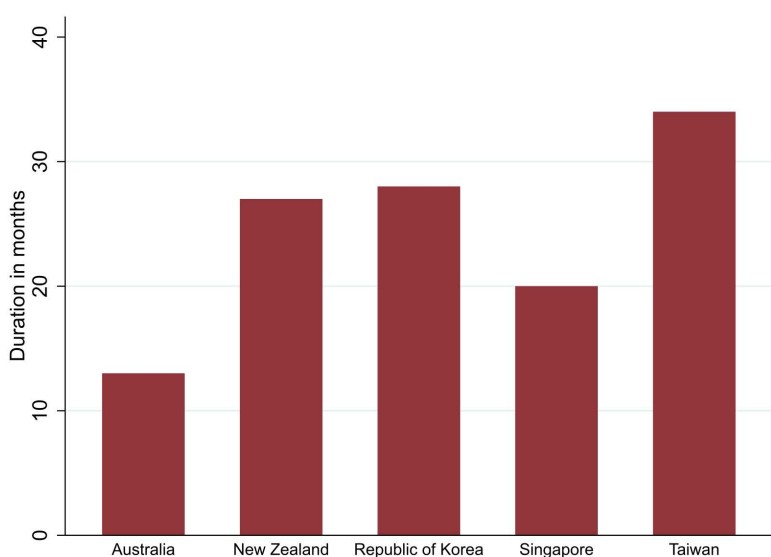

**Fig 2. Total months quarantine universally applied by the five countries (January 2020-May 2023).**

vaccination or recent infection with COVID-19). In addition, travellers deemed higher priority or "essential" were permitted reduced quarantine requirements including foreign students, family members, healthcare workers. Notably, all five countries encouraged international business travel from selected countries by reducing or exempting quarantine (subject to testing, vaccination or restricted itineraries).

Migrant workers were another category of traveller targeted by ITMs. Singapore suspended Work Passes for low-wage migrant workers from China from 31 January 2020, and then extended this suspension to other countries [58]. Foreign workers and other migrants in Singapore were not allowed to leave the country except for emergencies or compassionate reasons with approval from their employer and the Ministry of Manpower. When a worker shortage ensued for sectors reliant on migrant labour, such as construction and domestic service, limited numbers of migrant workers were permitted to enter Singapore, although subject to quarantine in designated facilities (e.g., approved hotels) similar to travellers from higher-risk countries. In March 2021, the Singaporean government established Migrant Worker Onboarding Centres (MWOC) to quarantine low-wage workers [59,60]. Similarly, Aotearoa New Zealand exempted Pacific Island seasonal workers from the Managed Isolation and Quarantine (MIQ) system in favour of quarantine in accommodation provided by employers.

International aircrew was another category of essential worker given special quarantine requirements. Given the impracticality of long quarantine periods for inbound crew, Australia and Taiwan required strict quarantining in designated accommodation until their outbound flights, usually a day or two afterwards. Aotearoa New Zealand allowed nationals and residents who were working aircrew to quarantine at home during layovers with the aim to maintain separation from domestic populations. If not departing again soon after, aircrew was subject to the same quarantine measures as other international arrivals. Singapore minimized layovers, with crew remaining onboard during turnaround. If overnight stops were necessary, aircrew were required to minimise contact with others including quarantine at an airport hotel [61].

Importantly, quarantine measures were focused on mitigating risk to domestic populations from international arrivals. While potentially infected outbound travellers posed a public health risk during a global pandemic, none of the countries required outbound travellers to quarantine before departure. Restrictions on outbound travel largely focused on reducing risks associated with return journeys. Singapore advised citizens and residents not to leave Singapore except for exigencies. Australians were not allowed to leave without permission from the government (which was regularly denied).

### Location of quarantine

There were two main locations where quarantine could be served in the five countries during the pandemic, with all using some combination of these (S5 Annex). The first was a location of choice, such as a residence, as long as contact with others was restricted. Sometimes referred to as "stay at home" or "self-quarantine," these sites were permitted in some countries because of the large number of inbound travellers to accommodate. During the initial phase of the pandemic, the five governments largely relied on self-quarantine given the limited availability of officially designated sites. While generally more convenient and less resource intensive, these locations needed to be monitored by public health officials to ensure compliance. Compliance mechanisms included apps, phone or video check-ins, and site visits. Location of choice quarantine was sometimes risk stratified, for example, by COVID-19 prevalence in the source country or the presence of non-travellers in the quarantine site.

The second location was facilities designated specifically for quarantine such as repurposed hotels or dormitories. These locations were closely supervised by one or more agencies (e.g., defense force, police) to ensure compliance. As COVID-19 cases increased worldwide and the risk of virus introduction from inbound travel increased, quarantine at designated supervised facilities became more common. Governments worked with hotel and dormitory operators to increase capacity [62]. In South Korea and Australia, arrivals who tested positive for COVID-19, or were symptomatic, were quarantined in facilities supervised by healthcare workers rather than military or police.

## Cost of quarantine and who paid

The cost of quarantine depended on where it was served (e.g., residence versus designated hotel), the standard of accommodation, additional costs covered (e.g., transportation, communications, food, testing, cleaning), and length of quarantine. Higher costs were associated with prolonged stays (up to 21 days) in designated sites of higher standard. Lower costs were incurred when quarantine was permitted in lower standard accommodation or place of residence (S6 Annex).

On who paid for these costs, in principle, States Parties to the IHR agree under Article 40 that "no charge shall be made by a State Party…for the following measures for the protection of public health" including quarantine. In practice, however, who paid varied by country and stage of the pandemic. During the initial phase, the five governments paid at least some costs. For example, repatriated nationals and permanent residents from high-risk locations were quarantined free of charge. Fees towards the cost of managed isolation and quarantine (MIQ) were not initially levied by Aotearoa New Zealand. South Korea did not charge international arrivals if they tested positive and needed to isolate and/or receive treatment. In Taiwan, all inbound travellers paid but citizens and residents could then apply for a subsidy.

As the pandemic continued, and quarantine was extended to more travellers, most governments sought to recover costs in whole or in part. South Korea, Singapore (from June 2020), and Taiwan required international arrivals to pay the full cost of quarantine. Singapore required employers to pay the cost of mandatory quarantine of temporary foreign workers in key sectors. In Australia, travellers were charged a contribution towards the cost of hotel quarantine. From August 2020, only nationals and visa class residents who travelled before the travel measures came into effect, and were returning and remaining home for a designated period (90 and then 180 days), were not charged in Aotearoa New Zealand. Two governments used cost to deter short term or non-essential travel such as tourism. Aotearoa New Zealand did not cover the costs of travellers staying for less than 90 (later extended to 120) days. Singapore did not cover the costs of nationals departing after a travel advisory was issued on 27 March 2020.

## Access to quarantine facilities

During the initial weeks of the pandemic, the five countries required repatriated nationals and residents returning from high-risk areas to stay in designated quarantine facilities. These travellers were limited in number. However, once the virus had spread worldwide, quarantine was extended to more categories of travellers, and quarantine became mandatory, the demand for designated quarantine facilities exceeded supply. The five countries then used different methods for providing access (S7 Annex).

To reduce traveller numbers, Australia and Aotearoa New Zealand quickly restricted inbound arrivals to nationals and residents. Repatriated Australian nationals and residents were initially quarantined on Christmas Island and then personal residences. From March 2020, quarantine was required in supervised facilities. To match the arrival numbers with available facilities, Australia capped permitted arrivals. Airlines were given the responsibility of staying within this cap when selling airline seats. By contrast, the Aotearoa New Zealand government allocated quarantine spaces randomly using an on-line ballot system, with a small number of spaces reserved for emergency or compassionate travel. A confirmed quarantine space was required before a flight could be booked. When the quarantine period was reduced in March 2022, the number of available spaces increased. Upon arrival in either Australia or Aotearoa New Zealand, travellers were randomly assigned to a specific quarantine facility. Nationals arriving in Taiwan, Singapore and South Korea could initially quarantine at home, while foreign nationals quarantined in designated facilities. As these countries shifted to designated facilities, demand was reduced by largely restricting international arrivals to nationals and permanent residents. In some cases, arrivals were randomly assigned accommodation either in a designated hotel or government facility. Despite this, supply was soon exceeded which was remedied by the establishment of quarantine hotels. Travellers could pre-book their own quarantine accommodation which gave them choice of location, quality and cost.

## Conditions of quarantine

In February 2020, WHO issued guidance to member states on the implementation of quarantine for COVID-19 containment. The guidance identified three key considerations: a) appropriate quarantine setting and adequate provisions; b) minimum infection prevention and control measures; and c) minimum requirements for health monitoring of quarantined persons. The following conditions were identified as "appropriate quarantine setting and adequate provisions":

- Adequately ventilated and spacious single rooms with ensuite toilet

- Suitable environmental infection controls, such as adequate air ventilation, filtration systems and waste-management protocols;

- Maintenance of social distancing (more than 1 metre);

- Accommodation with an appropriate level of comfort including:

  - Food, water and hygiene provisions;

  - Protection for baggage and other possessions;

  - Appropriate medical treatment for existing conditions;

  - communication in a language that they can understand explaining: their rights; provisions that will be made available to them; how long they will need to stay; what will happen if they get sick; contact information of their local embassy or consular support;

- Assistance for quarantined travellers, isolated or subject to medical examinations or other procedures for public health purposes;

- Assistance with communication with family members outside the quarantine facility;

- If possible, access to the internet, news and entertainment;

- Psychosocial support; and

- Special considerations for older individuals and individuals with comorbid conditions, due to their increased risk for severe COVID-19 disease [63].

The five countries, in turn, established national guidelines for minimum conditions in designated quarantine facilities. Assessment of the implementation of WHO and national guidelines has been limited to date. The few studies conducted have focused on selected populations or impacts [64], and do not separate domestic from travel-related quarantine requirements [65]. Moreover, comparative analysis is hindered by variations in standards of "good practice" across countries and thus lack of standardized data on common criteria.

As a starting point, we gathered available data on six criteria (S8 Annex): food (quality, sufficiency, appropriateness); facilities and amenities (accommodation, ventilation, sanitation and hygiene, communications, exercise, outdoor space); healthcare (pre-existing and COVID-related needs); special needs (disability or other accommodations); priority populations (children/parents, older adults); and cultural accommodations (food, spiritual, linguistic). In general, we found that basic needs were met in all five countries regarding food, facilities/amenities, and healthcare. In Singapore, Taiwan and South Korea, higher standards were available to those willing and able to pay. In Australia, some facilities chose to 'upsell' on arrival by, e.g., charging more for a room with a balcony or making a special menu available but this was executed outside of the official quarantine arrangements. Special needs, in the form of accessibility requirements, were not provided for in the five countries. Quarantine at home was permitted as an alternative in selected cases. The priority populations provided for were children/parents, and older adults. Limited attention was given to cultural accommodations, largely food

choices, but only for some cultural groups. Fuller understanding of what basic conditions should be met, if mass quarantine were needed in future, remains needed.

## Equity considerations in the use of quarantine in the five countries

Our findings suggest similarities among the five countries that led them to be considered exemplars during COVID-19. All were early adopters, applied quarantine relatively stringently (i.e., for most or all incoming travellers), maintained quarantine for most of the emergency phase of the pandemic, and required a long quarantine period (i.e., 14 days for much of the pandemic). This is seen to have supported the effective achievement of public health goals. However, our analysis also suggests differential impacts of quarantine implementation within and between the five countries. We describe these in relation to two aspects of equity: distribution of opportunity and distribution of burden. We recognize that other equity considerations arise when applying quarantine during public health emergencies that lay beyond the scope of this analysis.

### Distribution of opportunity

The ability to access quarantine facilities was an important determinant of whether employment, education, and personal relationship needs could be met during the pandemic. Equity considerations were raised by two aspects of how the distribution of opportunity to quarantine was managed. First, once quarantine became mandatory for inbound travellers, how did the five countries choose to increase supply or reduce demand? When demand for quarantine facilities exceeded supply, Singapore, South Korea and Taiwan managed this scarcity by allowing nationals and residents to quarantine at home. Designated facilities were reserved for foreign nationals. As the pandemic progressed, demand was reduced by limiting arrivals to nationals and residents. Location of quarantine was then based on public health risk assessment rather than availability of facilities. By contrast, Australia and Aotearoa New Zealand required quarantine to be undertaken in designated facilities only, and thus needed to move quickly to align demand with supply by restricting entry to nationals and residents, and operating quota systems.

Second, after managing the volume of international arrivals, how did the five countries allocate access to designated quarantine facilities as a scarce resource to inbound travellers? A range of allocative methods were available: first come, first served, market system (supply and demand regulated by price), means-testing, lottery, needs-based (those with greater need receive priority), or some combination of these. Equitable distribution of opportunity to access quarantine facilities is not simply equal opportunity, but opportunity relative to needs or circumstances. Our findings suggest some equity-promoting practices were adopted. For instance, Singapore provided migrant workers with designated quarantine facilities that could be booked by employers. In Aotearoa New Zealand, as demand outstripped supply, and waitlists became unmanageably long, a lottery system was introduced. The online system was reset each week, with would-be travellers randomly allowed to access the booking system until quarantine places for the week were filled. A limited number of spaces were available for emergency travel based on strict criteria. While high demand and limited supply meant that many citizens and residents could not travel home until ITMs were lifted, the lottery system created equitable distribution of opportunity for would-be returnees.

However, our findings suggest other allocative methods were equity-harming. In Australia, a cap on international arrivals (broken down by state/territory point of entry) was applied from July 2020 based on quarantine capacity. This cap was achieved by limiting flights to two daily services per point of entry, and operating flights at reduced capacity [66]. For the flights to be financially viable to airlines, 25 passengers were deemed by the industry to be the minimum threshold for a viable service, with seat prices significantly increased to business class fares [67]. Allocation of seats was thus determined by the market (ability to pay). The substantial cost of securing an airline seat, coupled with the expense of mandatory supervised quarantine (up to $3220 per adult), was not affordable to many. This, in turn, raised questions about fairness and other ethical concerns [36]. The permitted arrival of a small number of foreign nationals, under "business innovation and investment visas," and high-profile individuals who were allowed to self-quarantine in a location of their choice,

became a further source of resentment that the "rich and powerful cut the line" [68] while many Australians abroad could not afford to travel home.

### Distribution of burden

Distribution of burden concerns how the costs of quarantine are borne by different individuals and populations. These burdens will vary depending on how quarantine is implemented (described in this paper) and personal circumstances. Equity concerns, first, the extent these burdens were distributed fairly between individuals and society. In general, people who must quarantine bear burdens individually that benefit the wider community. Bioethicists argue that the principal of reciprocity "requires that the state has an obligation to assist individuals in discharging their duty to comply with public health measures and avoid disproportionate burdens accruing to populations or individuals." [69]. The IHR states that individuals should not bear the costs of quarantine for public health purposes [70]. As described above, the five governments initially covered the cost of quarantine requirements during the pandemic given the benefits to society. However, the significant cost of mass quarantine led governments to seek cost-recovery in whole or in part. This prompted different perspectives on how costs should be fairly distributed between individuals and society.

Second, equity concerns the extent these cost burdens were distributed fairly among individuals or population groups. The five countries sought to distribute burdens based on, for instance, category of traveller (e.g., citizenship and residence) or purpose of travel (e.g., essential versus discretionary). While foreign nationals generally faced restrictions from entering the five countries, where permitted, quarantine costs were levied. In New Zealand, foreign nationals deemed critical workers were liable for charges at a higher rate than citizens and visa class residents. In Singapore, the Ministry of Manpower directed employers and employment agencies to cover quarantine and associated costs such as testing and health insurance for migrant workers in key sectors such as construction and manufacturing.. In practice, these costs were generally transferred to low-wage migrant workers (average monthly salary was S$500 to S$600) as part of agency recruitment fees (ranging from S$5000 to S$12,000 prior to COVID-19) [71,72].

Some countries took account of financial hardship and the ability to pay, although these concessions were limited to citizens and selected residents. Aotearoa New Zealand provided the option of payment by instalments, and full or partial waivers based on hardship [73]. Australia permitted nationals and residents to request a fee waiver application based on hardship. However, applications could not be submitted until after quarantine ended and a bill was issued. Uncertainty about the criteria for fee waivers is reported to have contributed to financial and emotional stress [74,75]. In Taiwan, individuals facing financial hardship could apply for subsidies under the "Special Act for COVID-19 Prevention and Relief." Also, employers were responsible for migrant workers' quarantine expenses. Those who complied with regulations and received no salary or similar aid could apply for NT$1,000 per day in compensation from the Ministry of Health and Welfare in Taiwan [76].

In some countries, such as Singapore, there were efforts to ease costs and relieve pressure on managed sites by allowing quarantine at an alternative location such as a residence or second property. Taiwan also permitted the use of residence of relatives or friends. However, from an equity perspective, this privileged those who owned suitable properties (i.e., wealthier populations), while those without remained weresubject to higher costs. On travel costs, however, the burden was borne by each traveller with limited exception (e.g., emergency repatriation), As such, inequities of burden were created for individuals less able to afford such costs.

Beyond economic costs, fuller understanding of the distribution of other burdens remains needed. The 14-day quarantine period long maintained by the five countries, and the location and conditions of quarantine, were arguably more burdensome on some populations than others. Older adults, and people with young children, pre-existing conditions, mental health or other special needs, or caring responsibilities faced particular challenges during mandatory quarantine. While a one-size-fits-all approach to quarantine during COVID-19 sought to mitigate population-level health risks during a public health emergency, the need to provide accommodations to mitigate disproportionate burdens on such populations

merits further consideration. This need for greater equity in the distribution of burden has been recognized by reviews of the COVID-19 response in Aotearoa New Zealand [77], Australia [78], Singapore [79] and South Korea [80].

## Discussion

The unprecedented use of ITMs during the COVID-19 pandemic, and recognition of their potential need in future public health emergencies, supports fuller assessment of their effectiveness at advancing public health outcomes, alongside consideration of their secondary impacts. This paper addresses three knowledge gaps – the limited detailed study of quarantine use as an ITM, their secondary impacts or unintended consequences, and differences in implementation across national settings. Our findings raise several implications to inform future policy decisions.

First, our comparative analysis shows important similarities across the five countries that contributed to effective limitation of SARS-CoV-2 introduction and onward transmission via international travel at the initial stages of the pandemic. These findings broadly align with the Stringency Index of the Oxford Covid-19 Government Response Tracker (OxCGRT) project which uses nine metrics including "international travel controls" although does not specifically assess quarantine [69]. All were among the earliest adopters of quarantine requirements which remained continually in place in some form for most of the emergency period. The five countries also required most or all inbound travellers to quarantine, primarily in designated and supervised locations. Our findings thus provide a more detailed assessment of the relative stringency of quarantine use in the five countries, with Australia and New Zealand the most stringent.

Second, while all the countries are deemed exemplary from a public health perspective, differences in quarantine implementation were identified across the five countries, some associated with secondary impacts. Our findings provide new understanding of how these impacts were distributed within and across individuals and populations, inequities in the distribution of opportunity and burden, and the extent each of the countries sought to mitigate them. Inequities in the distribution of opportunities to quarantine were perhaps most significant in Australia where the cap on airline seats and quarantine places resulted in market-based costs that many could not afford. Aotearoa New Zealand allocated quarantine as a scarce resource through a lottery system, with emergency travel exemptions, to support fairness of access. The distribution of burden in the five countries also raised equity concerns regarding the appropriate balance between the cost to individuals and the benefits to society from quarantine. None of the five countries adhered to the principal of reciprocity, due to the unprecedented costs of mass quarantine, but efforts were made by Singapore, Taiwan and Korea to allow citizens to quarantine in their residences at certain points of the pandemic to ease financial burden. Some populations, such as people with children or special needs, were disproportionately burdened by how quarantine was implemented.

Third, amid the highly varied ways in which ITMs were applied by countries worldwide, our study offers detailed comparative analysis of five countries using a framework of eight key variables. This analysis enabled us to go beyond broad perceptions of exemplary practice to understand how different governments dealt with public health goals and secondary impacts. Our findings suggest difficult trade offs but also important opportunities to mitigate them through greater attention to equity considerations. For example, the disproportionate impacts of quarantine policies on low-wage migrant workers in Singapore led to changes in national policy including access to primary health care [81] and regulation of accommodation under the Foreign Employee Dormitory Act [82]. Given limitations in available data, our study could not provide systematic analysis of these inequities. There are also other forms of inequity, for example, related to procedural justice, that warrant fuller analysis. Yet the failure to address such concerns can lead to perceptions of unfairness that, in turn, have consequences for public trust in government [83] and undermine effective responses to future pandemic events.

## Conclusion

Evidence from the COVID-19 pandemic shows that varied types of ITMs may need to be used in future pandemic events to mitigate public health risks in a globalized world. Yet which measures should be effectively used, when, where and how remains poorly understood. In addition, the profound disruptions to individuals, societies and economies caused by travel

measures must be better addressed, notably when secondary impacts are inequitably distributed. This comparative analysis in five countries confirms practices that contribute to perceptions of their exemplary use of quarantine during COVID-19 to reduce virus introduction and onward transmission. However, quarantine also caused secondary impacts, including inequitable distributions of opportunity and burden, which were addressed differently in the five countries. We argue that the most exemplary countries were those that effectively navigated both public health goals and secondary impacts when applying quarantine measures.

## Supporting information

**S1 Annex. Legal framework for quarantine use during the COVID-19 pandemic in the five countries.**
(DOCX)

**S2 Annex. Timing of quarantine use by the five countries.**
(DOCX)

**S3 Annex. Length of quarantine period during the COVID-19 pandemic in the five countries.**
(DOCX)

**S4 Annex. Persons/populations subject to travel-related quarantine in the five countries.**
(DOCX)

**S5 Annex. Location of quarantine by international travellers in the five countries.**
(DOCX)

**S6 Annex. How much and who paid for quarantine in the five countries.**
(DOCX)

**S7 Annex. How quarantine facilities were accessed in the five countries.**
(DOCX)

**S8 Annex. Conditions of quarantine in designated facilities in the five countries.**
(DOCX)

## Author contributions

**Conceptualization:** Kelley Lee.

**Data curation:** Kelley Lee, Jane Williams, Yi-Chin Wu, Aysha Farwin, Youmi Kim, Sungkyu Lee, Natasha Howard, Salta Zhumatova.

**Formal analysis:** Kelley Lee, Jane Williams, Yi-Chin Wu, Aysha Farwin, Youmi Kim, Sungkyu Lee, Natasha Howard, Salta Zhumatova.

**Funding acquisition:** Kelley Lee.

**Investigation:** Kelley Lee, Jane Williams.

**Methodology:** Kelley Lee, Jane Williams.

**Validation:** Kelley Lee.

**Visualization:** Salta Zhumatova.

**Writing – original draft:** Kelley Lee, Jane Williams.

**Writing – review & editing:** Kelley Lee, Jane Williams, Yi-Chin Wu, Aysha Farwin, Youmi Kim, Sungkyu Lee, Natasha Howard, Salta Zhumatova.

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
