## [Decision Letter · Decision Letter 0]

12 Aug 2025

PGPH-D-25-01241

The use of quarantine as an international travel measure during the COVID-19 pandemic: A comparative analysis of implementation and equity impacts in five “exemplar” countries

Dear Dr. Lee,

Thank you for submitting your manuscript to PLOS Global Public Health. After careful consideration, we feel that it has merit but does not fully meet PLOS Global Public Health’s publication criteria as it currently stands. Therefore, we invite you to submit a revised version of the manuscript that addresses the points raised during the review process.

We look forward to receiving your revised manuscript.

Kind regards,

Bipin Adhikari, MBBS, DTM&H, MCTM, MPH, DPhil

Academic Editor

Journal Requirements:

1. Please note that PLOS Global Public Health has specific guidelines on code sharing for submissions in which author-generated code underpins the findings in the manuscript. In these cases, all author-generated code must be made available without restrictions upon publication of the work. Please review our guidelines at https://journals.plos.org/globalpublichealth/s/materials-and-software-sharing#loc-sharing-code and ensure that your code is shared in a way that follows best practice and facilitates reproducibility and reuse.

i. State the initials, alongside each funding source, of each author to receive each grant.

ii. State what role the funders took in the study. If the funders had no role in your study, please state: “The funders had no role in study design, data collection and analysis, decision to publish, or preparation of the manuscript.”

3. Please send a completed 'Competing Interests' statement, including any COIs declared by your co-authors. If you have no competing interests to declare, please state "The authors have declared that no competing interests exist". Otherwise please declare all competing interests beginning with the statement "I have read the journal's policy and the authors of this manuscript have the following competing interests:"

4. Please ensure that your Ethics Statement is available in its entirety at the beginning of your Methods section, under a subheading 'Ethics Statement'. It must include:

1) The name(s) of the Institutional Review Board(s) or Ethics Committee(s)

2) The approval number(s), or a statement that approval was granted by the named board(s) 

3) (for human participants/donors) - A statement that formal consent was obtained (must state whether verbal/written) OR the reason consent was not obtained (e.g. anonymity). 

NOTE: If child participants, the statement must declare that formal consent was obtained from the parent/guardian.

5. Please provide separate figure files in .tif or .eps format.

6. We note that your Data Availability Statement is currently as follows: “All data used in this paper, compiled from publicly available sources, are included in the manuscript. The WHO Public Health and Social Measures dataset is no longer available on the WHO website. The data used to create the two figures for this paper can be provided either as an additional on-line document or posted on the Pandemics and Borders Project website. We welcome guidance from the editors on where best to situate this data.”

Additional Editor Comments (if provided):

Reviewers' comments:

Reviewer's Responses to Questions

**Comments to the Author**

1. Does this manuscript meet PLOS Global Public Health’s publication criteria?

Reviewer #1: Yes

Reviewer #2: Yes

Reviewer #3: Partly

2. Has the statistical analysis been performed appropriately and rigorously?

Reviewer #1: Yes

Reviewer #2: N/A

Reviewer #3: N/A

3. Have the authors made all data underlying the findings in their manuscript fully available (please refer to the Data Availability Statement at the start of the manuscript PDF file)?

Reviewer #1: Yes

Reviewer #2: No

Reviewer #3: Yes

4. Is the manuscript presented in an intelligible fashion and written in standard English?

Reviewer #1: Yes

Reviewer #2: No

Reviewer #3: Yes

Reviewer #1: This is an insightful article and a much-appreciated review of quarantine practices during the pandemic. Would suggest you consider the criteria listed from SARS in Toronto - https://pmc.ncbi.nlm.nih.gov/articles/PMC2094974/

Great inclusion of payments per quarantine. Would suggest you lightly touch on what time of government/political system is in place within each country.

What about in-country travel across territories? Thinking of NSW vs QLD, etc. Realizing this is focused on intl travel, but might be worth mentioning. Would also consider noting that the reliance on such significant quarantine measures often impacted other response - such as the rollout of vaccines (thinking Aus), which then allowed for significant outbreaks (e.g. Sydney cruiseship) and false sense of security.

Reviewer #2: This paper presents detailed comparison of quarantine use in the five exemplary countries based on the 8 components in the heuristic framework for comparative analysis and the equity framework focusing on cost and benefit sharing. It highlights how various degree of quarantine measures impacts on effective containment of virus along with examples of cost-sharing mechanisms practiced by the exemplary countries.

The paper has an excellent start with the information on international travel measures (ITM) and its role on achieving public health goals during pandemic. However, the introductory paragraphs could be rearranged under single heading, Introduction, to establish logical and improve readability as: Concept of ITM, quarantine (page 16-17, conditions of quarantine), and isolation (Page 4, 1st and 2nd para.), equity lens (Page 7, 3rd. para.), literature summary (Table 1), evidence gap (Page 3, 2nd. and 3rd para.), and objectives. It is suggested to highlight the existing evidence with appropriate paraphrasing and citation rather than mentioning it with “….” (for instance, Page 5, line 7-13).

Additionally, the last paragraph of the introduction section (Page 4) might be more appropriate in the approach and methods section, as it delves into methodological details. It is suggested to differentiate primary and secondary data mentioned on Page 6, line 1, followed by case study selection, frameworks, data collection tools and coding of quarantine practices. I am curious about the 0-1 scale used for quarantine policy (Page 7, line 1-3). Moreover, the paragraphs explaining equity lens, on equity framework (Page 7-8) is insightful. However, considering the section where it is placed, it can be made more concise, limiting to the type of justice and relevant questions. A table with detailed information can be added on supplementary file for reference.

The depth of information provided in the result section, under the components (legal mandate, timing, duration, population, location, cost, access, conditions) of the heuristic framework for comparative analysis of quarantine implementation and equity framework is commendable. However, as a reader, I felt it would more appropriate to discuss the first paragraph (page 8-9) in Introduction/Discussion section. Moreover, a few sentences seemed to lack the needful information as well. [For instance, page 8, last para., line 2- included in which implementation framework? Page 10, 1st. para, line 1- which 4 countries embedded quarantine within broader legislation? And line 2- which three countries made temporary expansion ?] Page 10, last para., line 7: can we mention the word “perhaps” in the result section ?

Further, it is suggested to insert footnotes for terms like travel bubbles, travel corridors (page 11, 2nd para).

Further, I felt there are few paragraphs in the result section which has better scope in the discussion section of the manuscript. For instance, page 17 last para. Similarly, the findings on page 20, 2nd para is an interesting one but this section requires logical flow and connection between the paragraphs. Else, it can be shifted in discussion section as well.

The discussion section still has more areas of the findings yet to be discussed. The findings from heuristic framework and equity framework can be discussed to conclude on the varying degree of public health impacts of those components in five exemplar countries. Similarly, it can be linked with significance of these findings at the end.

Reviewer #3: Comments:

Authors' efforts in bringing together this important piece of evidence are commendable. The manuscript presents an important review and documentation on the use of quarantine as an international travel measure during the COVID-19 pandemic, with a comparative analysis of its implementation and equity impacts across five “exemplar” countries. The review is relevant; however, the manuscript is too lengthy and would benefit from concise revisions to improve overall readability and clarity.

Specific comments:

1. The manuscript is too long. Consider making it more concise to enhance readability and alignment with journal guidelines.

2. Several tables and supplementary information can be moved to an appendix or online supplementary materials.

3. Abstract: The abstract lacks clarity, particularly in describing the methodology and key findings. The type of study, Data sources, Key results, and Main conclusions should be clearly stated.

4. Introduction and Background: The distinction between the Introduction and Background is unclear. Please refer to the journal’s author guidelines: if a separate background section is required, ensure it has a distinct focus; otherwise, consider merging it with the introduction. The “Background” section title may be changed to something more specific or merged as needed.

5. Methods: The methods section lacks detail. What specific methodology was used? While secondary data use is mentioned, the process of data collection, analysis framework, and ethical considerations needs to be stated clearly. Clarify the study design. What type of review is this? Terms such as “exemplar countries” and “eight variables” should be briefly clarified. What criteria defined “exemplar” status?

6. Results: The presentation of results is too descriptive in places. Data should be referenced and dated precisely (e.g., periods of quarantine, policies used). The condition of quarantine must be compared to standard guidance or relevant country-specific policies, including references.

7. Page 47: The photo included needs a source and context. Where was this taken? Is permission required?

8. Tables 3–9 are too long and hinder the manuscript’s flow. Consider moving some to supplementary files.

9. Tables 4, 5, 6, 7, 8, and 9 should be made more concise and reader-friendly, or can be moved to supplementary files.

10. Discussion and Conclusion: Ensure the discussion reflects how the findings contribute to the current knowledge. Link findings more directly to the literature and to policy/practice implications. Keep the conclusion concise, and ensure it clearly summarises the key takeaways.

**Do you want your identity to be public for this peer review?** For information about this choice, including consent withdrawal, please see our Privacy Policy

Reviewer #1: No

Reviewer #2: No

Reviewer #3: **Yes: ** Devendra Singh

---

## [Decision Letter · Decision Letter 1]

27 Oct 2025

The use of quarantine as an international travel measure during the COVID-19 pandemic: A comparative analysis of implementation and equity impacts in five “exemplar” countries

PGPH-D-25-01241R1

Dear Professor Lee,

We are pleased to inform you that your manuscript 'The use of quarantine as an international travel measure during the COVID-19 pandemic: A comparative analysis of implementation and equity impacts in five “exemplar” countries' has been provisionally accepted for publication in PLOS Global Public Health.

Best regards,

Bipin Adhikari, MBBS, DTM&H, MCTM, MPH, DPhil

Academic Editor

Reviewer Comments (if any, and for reference):

Reviewer's Responses to Questions

**Comments to the Author**

Reviewer #2: All comments have been addressed

Reviewer #3: All comments have been addressed

publication criteria?

Reviewer #2: Yes

Reviewer #3: Yes

3. Has the statistical analysis been performed appropriately and rigorously?

Reviewer #2: Yes

Reviewer #3: Yes

4. Have the authors made all data underlying the findings in their manuscript fully available (please refer to the Data Availability Statement at the start of the manuscript PDF file)?

Reviewer #2: Yes

Reviewer #3: Yes

5. Is the manuscript presented in an intelligible fashion and written in standard English?

Reviewer #2: Yes

Reviewer #3: Yes

Reviewer #2: The manuscript seems to lot more concise and readable than the prior version. Page 5: Are you referring to Table 1 there (cannot find Table 2 attached)? I still think the discussion part can be written in more detail. The comparison and practices of exemplar countries can be discussed in depth with some author's interpretations.

Reviewer #3: Comments has been addressed.

**Do you want your identity to be public for this peer review?** For information about this choice, including consent withdrawal, please see our Privacy Policy

Reviewer #2: No

Reviewer #3: No
